# CRISPR/Cas9-Mediated Mutagenesis of *Antennapedia* in *Spodoptera frugiperda*

**DOI:** 10.3390/insects15010016

**Published:** 2023-12-29

**Authors:** Congke Wang, Te Zhao, Xiaolong Liu, Tianliang Li, Leiming He, Qinqin Wang, Li Wang, Lin Zhou

**Affiliations:** 1College of Plant Protection, Henan Agricultural University, Zhengzhou 450046, China; 2Key Laboratory of New Pesticide Development and Application, Henan Agricultural University, Zhengzhou 450046, China; 3Green Pesticide Creation Engineering Technology Research Center, Henan Agricultural University, Zhengzhou 450046, China

**Keywords:** *Spodoptera frugiperda*, *Antp*, CRISPR/Cas9, thoracic segment development, thoracic leg

## Abstract

**Simple Summary:**

*Spodoptera frugiperda*, a significant pest affecting various crops, has been the focus of study to understand the role of *Antennapedia* (*Antp*), essential for insect thorax and wing development. We investigated *Antp* in *S. frugiperda* using RT-qPCR and CRISPR/Cas9 genome editing. We found that *Antp* is highly similar across Lepidoptera and is expressed throughout the life cycle of *S. frugiperda*, with the highest expression in the egg stage and significant expression between 12 to 48 h. The gene was primarily active in the thorax and legs. By knocking out *Antp* by CRISPR/Cas9, we observed abnormal development of the thoracic legs of the larvae and abnormal pupation. After the expression of *Antp* decreased, the expression of other Hox genes, appendage development genes and cuticular protein genes decreased significantly.

**Abstract:**

The homeotic gene *Antennapedia* (*Antp*) has been identified as playing a pivotal role in the morphogenesis of the thorax and wings across various insect species. Leveraging insights from previous studies, the functional characterization of *Antp* in *S. frugiperda* was undertaken using RT-qPCR and the CRISPR/Cas9 genome-editing system. Phylogenetic analyses indicate that *Antp* shares a high degree of sequence homology among Lepidoptera species. The expression profile of *SfAntp* was detected by RT-qPCR. The results showed that *SfAntp* was expressed in the whole growth cycle of *S. frugiperda*, the expression level was the highest in the egg stage, and the expression level was higher from 12 h to 48 h. Tissue-specific expression profiling demonstrated that *SfAntp* was most abundantly expressed in the thoracic segments and legs. To functionally disrupt *SfAntp*, two sgRNA sites were designed at the first exon of *SfAntp* and the gene was knocked out by CRISPR/Cas9 via microinjection. The results showed that the deletion of *SfAntp* produced a mutant phenotype of thoracic fusion, thoracic leg defect, leg-like protrusions between the head and thoracic segments and pupation deformity. In addition, deletion of *SfAntp* resulted in high embryo mortality. Through DNA sequencing, it was found that the target site of the *SfAntp* mutant had different degrees of frameshift mutations, indicating that the mutant phenotype was indeed caused by the knockout of *SfAntp*.

## 1. Introduction

*Spodoptera frugiperda* (J. E. Mith) belongs to Lepidoptera, Noctuidae, also known as fall armyworm [1]. It is native to tropical and subtropical regions of the Americas and has been marked by the Food and Agriculture Organization (FAO) as a major global migratory pest [2,3]. It has characteristics such as fast migration speed, wide harm range, many host species and strong reproductive ability. *S. frugiperda* has now spread globally, in Africa, Asia, Europe, Oceania and other regions, and in January 2019, it spread rapidly in China’s Yunnan region and the surrounding provinces [4,5,6]. Now, *S. frugiperda* is still in a state of continuous outbreak in China [7].

*Antp* is a member of the *Drosophila* Antennapedia Complex (ANT-C) and is the earliest discovered Hox gene in *Drosophila* [8]. In 1915, a class of homologous heteromorphic mutations caused by *Antp* was found in *Drosophila*, which led to the phenotypic transformation of cranial antennae into thoracic legs, thus bestowing the designation “Antennapedia” [9]. In *Drosophila*, *Antp* expression commences during the nascent phases of embryonic development, predominantly within the thoracic region [10]. Initial expression unfolds from the posterior labial segment, pervading through the thoracic cavity, and into the first abdominal segment. As embryonic development progresses with concurrent shortening of the embryo, the expression domain becomes confined predominantly to the T1–T3 thoracic segments. This spatial restriction of expression may arise from the negative regulatory influences exerted by the *Ultrabithorax* (*Ubx*) [11]. *Antp* expression is tightly regulated during development, with two promoters (P1 and P2) targeting mid-thoracic tissue, each influenced by specific enhancers active in early embryogenesis, and various mechanisms modulating larval tissue expression [12].

In *Drosophila*, *Antp* plays a pivotal role in governing cell fate determination within the head and thoracic regions, somite differentiation, and the morphogenesis of tissues and organs [13]. Mutations in *Antp* give rise to an array of phenotypic manifestations. Specifically, the loss of *Antp* function in the embryo culminates in a homeotic transformation, such that the identities of the second and third thoracic segments are converted to that of the first thoracic segment [14]. Moreover, in imaginal discs of adult *Drosophila*, silencing of *Antp* induces the metamorphosis of central thoracic appendages into antennal structures [15]. Although the underlying mechanisms for these distinct phenotypic alterations remain enigmatic, they likely reflect the intricately specialized developmental stages of *Drosophila* embryos and larvae. *Antp*, together with other Hox proteins, is crucial for the proper development of embryonic musculature and subsequently contributes to the architecture of the adult thoracic musculature [16]. Furthermore, *Antp* is instrumental in the development of the visual system, nervous system, and midgut, underscoring its significance in the comprehensive development of *Drosophila* [17,18].

Recently, a novel type of *Wedge eye-spot* (*Wes*) mutation has been identified in *Bombyx mori*. Homozygous larvae display eye patterns with a wedge shape, thoracic fusion, and antennae-like appendages positioned between the head and thorax. The occurrence of these phenotypes can potentially be attributed to the loss of *Antp* functionality [19]. *Antp* is not necessary for wing development in *Drosophila*, and the silencing or overexpression of this gene will not affect the morphology of *Drosophila* wings [20]. However, in the *Wes* mutant of *B. mori*, both the fore and hind wings exhibit smaller or deformed structures, and the expression levels of *Antp* in these mutant wings are significantly higher compared to normal wings. These observations suggest that the role of *Antp* may differ across various insect species.

Previous investigations on *Tribolium castaneum* have revealed that mutations in *ptl*, an *Antp* homolog, can induce the transformation of three pairs of thoracic legs into antennae [21]. Similarly, disrupting *Antp* in the spider *Achaearanea tepidariorum* leads to the production of supernumerary legs in the thoracic region [22]. Notably, *Dll* represents a pivotal gene involved in leg development in both fruit flies and spiders [23,24]. In *Tetranychus urticae*, *Scr*, *Ftz*, and *Antp* exhibit significant upregulation during the first molt stage. RNA interference results in the curling of L3 and loss of L4, while these three genes also collaborate in maintaining leg development in *T. urticae* alongside *Dll* [25]. However, after interfering with *Antp* in spiders, the expression of *Dll* at the leg growth site does not show downregulation. This discrepancy may arise from the interaction of *Antp* with different cooperative factors and target genes during the course of species evolution [22].

Hox genes play a crucial role in the development and evolution of insect appendages [26]. Each Hox gene regulates the expression of a multitude of target genes and plays a pivotal role in the formation of the insect’s body axis, nervous system, and organs [27]. *Antp* is related to the development of insect thoracic segments and the formation of thoracic legs and plays a key role in the embryonic development of insects [28,29,30]. In view of its importance, this study carried out bioinformatics analysis, spatiotemporal expression pattern analysis and CRISPR/Cas9 mutagenesis of the *S. frugiperda Antp* (*SfAntp*) gene. The purpose is to obtain the somatic development mutants of different parts from the mutation of this Hox gene in *S. frugiperda*, and to explore its role in the development of appendages of *S. frugiperda*, so as to lay a theoretical basis for the population genetic control of *S. frugiperda*.

## 2. Materials and Methods

### 2.1. Insects

*S. frugiperda* used in this experiment were purchased from Henan Jiyuan Baiyun Industrial Co., Ltd (Jiyuan, China). They were reared in an artificial climate chamber (temperature 28 ± 1 °C, photoperiod L:D = 14:10, relative humidity 75%). The larvae were fed with fresh corn leaves planted in the laboratory, and the adults were fed with 10% sucrose solution for nutrition.

### 2.2. Phylogenetic Tree

We selected 28 *Antp* sequences on NCBI to align with *SfAntp* and establish a phylogenetic tree. Sequence alignment was constructed by using the maximum likelihood method based on CLUSTAL W2. The neighbor-joining method was used to create the tree from 29 available *Antp* sequences. The evolutionary distances were computed using the Poisson correction method and are displayed as the number of nucleotide substitutions per site.

### 2.3. Analysis of SfAntp Expression Profile

The analysis of expression profiles at different developmental stages required RNA extraction from samples, including eggs, larvae of stage 1–6, female and male pupae, and female and male adults.

Furthermore, for the expression profile of embryonic development, it was necessary to collect the eggs at 6 h, 12 h, 24 h, 36 h, 48 h and 60 h after oviposition for RNA extraction.

In addition, the expression profile analysis of different tissues required the extraction of RNA from different tissues of male and female adults, including head, thoracic segments, abdominal segments, external genitalia, legs, antennae, and wings.

RNA was extracted with the Trizol method; cDNA was synthesized by reverse transcription and stored at −80 °C.

The sequence of *SfAntp* (XM_035584826) was obtained from NCBI, and the primers for RT-qPCR were designed by Primer software (version 5). The specific sequences are shown in Table 1. The primers used for RT-qPCR were first tested for primer amplification efficiency. A series of diluted cDNAs (1×, 10×, 100×, 1000×, 10,000×) were used to construct a standard curve to calculate its correlation coefficient and slope value. The amplification efficiency was calculated with the equation [10(1/−slope) − 1] × 100%. The amplification efficiency needs to be between 90 and 110% before the next experiment can be performed.

RT-qPCR was performed on a 20 μL system. The content of each reaction component was 10 μL of Power Up^TM^ SYBR^TM^ Green Master Mix, 1 μg of cDNA, 1 μL of 10 μM forward and reverse primers each, plus RNase Free ddH_2_O to 20 μL. The reaction conditions were 50 °C for 2 min, 95 °C for 2 min; 40 × (95 °C for 15 s, 60 °C for 1 min); 95 °C for 15 s, 60 °C for 1 min to 95 °C in 0.15 °C steps of 1 s for the dissolution curve. GAPDH was used as a reference gene, and three replicates were set for each sample. After the reaction, the collected data were analyzed by using the 2^−ΔΔCt^ method in Excel.

### 2.4. Phenotypic Changes Caused by SfAntp Mutation

In this study, the first exon of *SfAntp* was selected as the target, and two SgRNAs were designed on it. The synthesized *SfAntp*-sgRNA1 (300 ng/μL), *SfAntp*-sgRNA2 (300 ng/μL) and Cas9 (600 ng/μL) proteins were mixed together. The eggs oviposited within 2 h were selected and injected with InjectMan 4 of Eppendorf. A biological repetition includes three technical repetitions. In each technical replicate, 147 eggs in the treatment group (KO-*Antp*) and 100 eggs in the control group (ddH_2_O) were used. Three biological replicates were performed in batches. After the injection was completed, the eggs were collected into a Petri dish, and a suitable amount of fresh corn leaves were placed in the Petri dish to moisturize, and the Petri dish was placed in a 27 °C incubator. The hatching rate was counted three days after injection. One-way analysis of variance and Tukey’s multiple test were performed on the hatching rates of the three biological replicates (*p* < 0.05). The larvae phenotype was observed and photographed. The mutant DNA was extracted according to the instructions of TIANGEN TIANamp Genomic DNA Kit. The specific amplification primers *SfAntp*-F and *SfAntp*-R containing the target were used as a template for PCR and then sequenced.

### 2.5. Expression Changes of SfAntp Mutant Related Genes

A mutant was randomly selected, RNA was extracted using the Trizol method, cDNA was synthesized by reverse transcription, and stored at −80 °C. We searched NCBI for sequences of other Hox genes (*Lab*, *Pb*, *Dfd*, *Scr*, *Ubx*, *Abd-A*, *Abd-B*) and used Primer software (version 5) to design RT-qPCR primers (Table 1).

## 3. Results

### 3.1. Phylogenetic Analysis of SfAntp

The coding sequence of *SfAntp* was 912 bases. Phylogenetic analysis based on *Antp* sequences of 28 other insects showed that *SfAntp* was clustered with homologs of other Lepidoptera species and had the closest relationship with *Spodoptera litura* and *Trichoplusia ni*, suggesting that they were structurally conserved (Figure 1).

### 3.2. SfAntp Expression Profile Analysis

#### 3.2.1. Expression Profile Analysis of *SfAntp* at Different Developmental Stages

*SfAntp* was expressed in the whole growth cycle of *S. frugiperda*; the expression level was the highest in the embryonic stage, while the expression level was lower in the larval, pupal and adult stages (Figure 2A). In order to speculate the possible role of *SfAntp* in the embryonic stage, total RNA was extracted from embryos at different developmental stages for RT-qPCR analysis. The results showed that the relative expression of *SfAntp* was the lowest at 6 h post oviposition (hpo), and the relative expression was higher at 12–48 hpo (Figure 2B).

#### 3.2.2. Expression Analysis of *SfAntp* in Different Tissues of Male and Female Adults

*SfAntp* was expressed in all tissues of *S. frugiperda*. The expression level of *SfAntp* was the highest in the leg and thorax of the female adult, while the expression level was lower in the wing, head, antennae, abdominal and external genitalia (Figure 2C). The expression of *SfAntp* in the male adult was similar to that in the female adult (Figure 2D).

### 3.3. Phenotypic Changes Caused by SfAntp Mutation

The hatching rate was 15.19% in the treatment group and 67.67% in the control group 3 days after injection (Table 2). There is a significant embryonic lethality after the knockout, suggesting that *Antp* is essential for embryonic development of *S. frugiperda*.

The results of three biological replicates’ data analysis showed no significant difference, so the hatching rate was expressed as the average of three biological replicates.

In this study, *SfAntp* was knocked out by CRISPR/Cas9. In order to identify that the mutant phenotype was caused by injection of *SfAntp*-sgRNA and Cas9 protein, five mutants were randomly selected to extract genomic DNA. After DNA purification, it was ligated with the vector and transferred into competent cells for ampicillin screening. The genotype of the mutants was verified by monoclonal sequencing. The results showed that two mutants had different degrees of frameshift mutations at both targets, and three mutants had a frameshift mutation occurring in one target (Figure 3A,B).

The wild-type *S. frugiperda* larvae have 1 head segment, 3 thoracic segments (T1–T3) and 10 abdominal segments (A1–A10). There are three pairs of thoracic legs on T1–T3 (Figure 4A), four pairs of abdominal legs on A3–A6, and one pair of tail legs on A10 in larvae. After *SfAntp* was knocked out, the development of the thoracic legs of *S. frugiperda* was significantly affected. At the larval stage, the phenotypes of less thoracic legs, thoracic leg fusion, the protuberance structure between the head and the thoracic segments, fusion of the head and the thoracic segments, and thoracic leg development defects were produced (Figure 4B–F). Most of the mutant larvae that could hatch normally died before the fifth instar, and only the larvae with one T2 segment lacking one thoracic leg survived to the pupal stage, which then showed abnormal leg and wing development at the corresponding position of the pupal stage (Figure 4G). In addition, some *S. frugiperda* which did not show obvious mutation in the larval stage showed structural abnormalities when they developed to pupal stage, such as the absence and abnormal development of the corresponding positions of the thoracic leg and wing, incision-like segments in A3–A5 abdominal segments, and the preservation of leg-like projections in A5–A7 abdominal segments (Figure 5B–E). These results indicate that *Antp* has an important effect on the development of thoracic tissue during the development of *S. frugiperda*, and the deletion will lead to mutations in this part.

### 3.4. Expression Changes of SfAntp Mutant Related Genes

In order to understand how *Antp* plays a regulatory role in the Hox gene family, a pupal mutant was randomly selected and the relative expression of the Hox gene family was analyzed by RT-qPCR. The knockout efficiency of *SfAntp* was verified first, and the results showed that its expression level decreased significantly (Figure 6A). Then the expression of related genes was detected, Hox genes were downregulated to varying degrees, among which *Scr*, *Ubx*, *Abd-A* and *Abd-B* were most significantly downregulated (Figure 6B). At the same time, according to the effect of *SfAntp* on the leg development of *S. frugiperda*, the relative expression levels of four genes related to leg development and two cuticular protein genes were detected. The results showed that the leg development related genes *Dac*, *Dll*, *Hth*, *Exd* and the cuticular protein genes *CPG24*, *CPG9* were significantly downregulated (Figure 6C).

## 4. Discussion

Hox genes are pivotal in the morphogenesis and evolutionary diversification of insect appendages. In the current investigation, we focused on *Antp* from the Hox gene family. Employing a suite of methodologies, including phylogenetic analysis, RT-qPCR, and the CRISPR/Cas9 genome-editing system, we meticulously examined the temporal and spatial expression patterns of *SfAntp* throughout the developmental stages of *S. frugiperda*. Additionally, we assessed the impact of *SfAntp* on the developmental processes in *S. frugiperda*.

The expression dynamics of *SfAntp* throughout the ontogeny of *S. frugiperda* were interrogated using RT-qPCR. The assay elucidated that *SfAntp* transcript levels peaked during the oogenesis, implying its primary function during early embryogenesis. Therefore, this experiment also detected the expression of *SfAntp* in different stages of embryonic development. It was found that *SfAntp* had a higher expression level between 12 h and 48 h of embryonic development. At the same time, the relative expression of *SfAntp* in different tissues of male and female adults was also detected. The results showed that *SfAntp* was highly expressed in the thoracic segments and legs of male and female adults, indicating that *Antp* was indeed involved in the regulation of thoracic segment and thoracic leg development.

To further investigate the functional role of *SfAntp*, two distinct sgRNA target sites were engineered within the first exon of *SfAntp*. Utilizing microinjection techniques, Cas9 protein in conjunction with the designed sgRNAs were introduced into *S. frugiperda* embryos. This targeted gene disruption of *SfAntp* precipitated a premature cessation of embryonic development, manifesting as increased embryonic lethality. A scant number of embryos successfully hatched. However, none of these specimens was viable to maturity, underscoring the critical function of *SfAntp* throughout the life cycle, particularly during embryogenesis and larval development in *S. frugiperda*.

The experimental disruption of *SfAntp* resulted in a constellation of mutant characteristics, including the fusion of thoracic segments, impairments of the thoracic appendages, and anomalous protuberances reminiscent of legs between the head and thoracic segments. These phenotypic alterations in *S. frugiperda* mirror those previously documented in *B. mori*, where *Antp* loss-of-function mutations prompted comparable developmental aberrations: thoracic segment fusion, incomplete thoracic and appendage morphogenesis, and the emergence of antenna-like structures interfacing the head with the thorax [19]. Phylogenetic analyses further corroborated the closely allied nature of *SfAntp* to other lepidopteran species, intimating the putative conservation of *Antp*’s regulatory sphere concerning thoracic segmentation and tissue development within lepidopteran insects. Moreover, peculiarities were observed in the mutagenized pupae, such as incision-like indentations in the A3–A5 abdominal segments and persistent larval leg-like protuberances within the A5–A7 regions, suggesting *Antp*’s role in the larval–pupal metamorphosis. In addition, in *Drosophila*, inhibition of the function of *Antp* will lead to a homologous transformation from the mid-posterior thorax to the anterior thorax, and *Antp* is closely related to the morphological determination of the thoracic leg [14]. These insights collectively endorse the concept of a conserved regulatory mechanism orchestrated by *Antp* across insect thoracic development.

*Antp* function extends beyond thoracic development; it is implicated in influencing other complex structures such as wings and eyespots in various Lepidoptera. In *B. mori*, *Antp* mutations have led to compromised wing development, resulting in smaller or malformed wings [31]. Similarly, *Antp* has been identified as crucial for the formation of butterfly eyespots, specifically in *Bicyclus anynana*, where it contributes to eyespot patterning on the forewings and determines the size of those on the hindwings [32]. In this study, *SfAntp* mutants of *S. frugiperda* never reached adulthood due to lethality at larval or pupal stages, rendering it impossible to directly observe adult wing formation. However, signs of abnormal wing development could be discerned in the pupae, paralleling the wing development issues seen in *B. mori*. The expression patterns of *SfAntp* in wild-type *S. frugiperda* suggest that although the gene is expressed in adult wings, its expression is more pronounced in the thorax and legs. This expression profile hints at a more dominant role of *SfAntp* in thoracic and appendage development compared to wings in *S. frugiperda*. This differential expression suggests that while *SfAntp* is important for wing development, its regulatory functions are more crucial or perhaps more complex in the development of the thorax and legs. Further research is warranted to fully elucidate the specific regulatory mechanisms of *SfAntp* within these various tissues and developmental stages.

## 5. Conclusions

The study involving CRISPR/Cas9-mediated knockout of *SfAntp* at the embryonic stage in *S. frugiperda* has yielded significant insights. The knockout manifestly induced embryonic lethality, suggesting that *SfAntp* is crucial for initial developmental processes. Besides hindering normal development and pupation, the mutation concurrently suppressed the expression of additional Hox genes, elucidating a possible regulatory network wherein *SfAntp* influences or is integrated with the broader Hox gene cascade. The persistence of larval leg-like protrusions in the abdominal segments of mutant pupae infers a specific role for *SfAntp* in the metamorphic process, seemingly promoting the transition from larval to adult stages. These results provide candidate genes for genetic control of Lepidoptera pests such as *S. frugiperda*.

## Figures and Tables

**Figure 1 insects-15-00016-f001:**
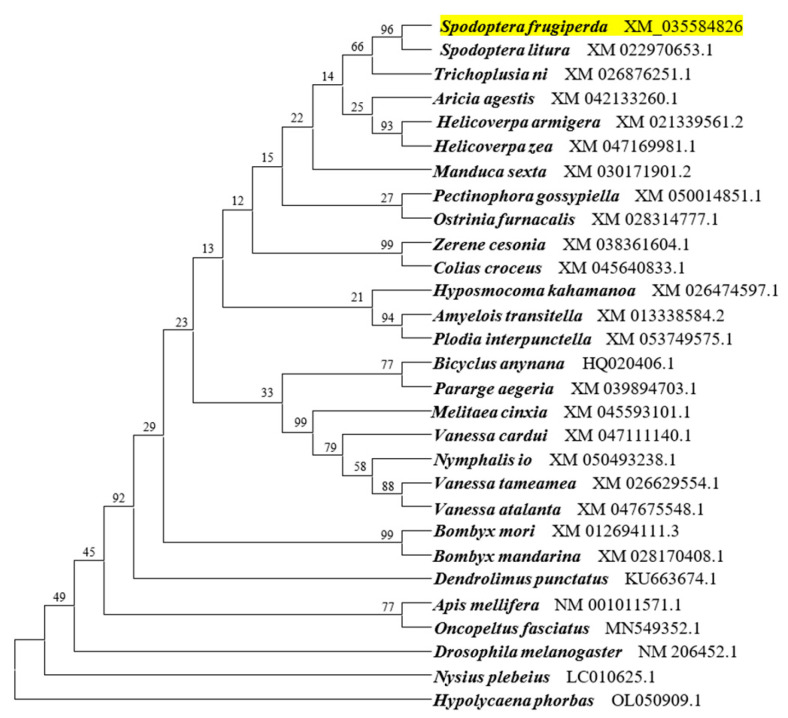
Phylogenetic analysis of *SfAntp*. The evolutionary histories of *Antp* were inferred using the neighbor-joining method. The percentages of replicate trees in which the associated taxa clustered together in the bootstrap test (1000 replicates) are shown next to the branches. Evolutionary distances were computed using the Poisson correction method and are listed as the number of nucleotide substitutions per site. *SfAntp* was highlighted in yellow.

**Figure 2 insects-15-00016-f002:**
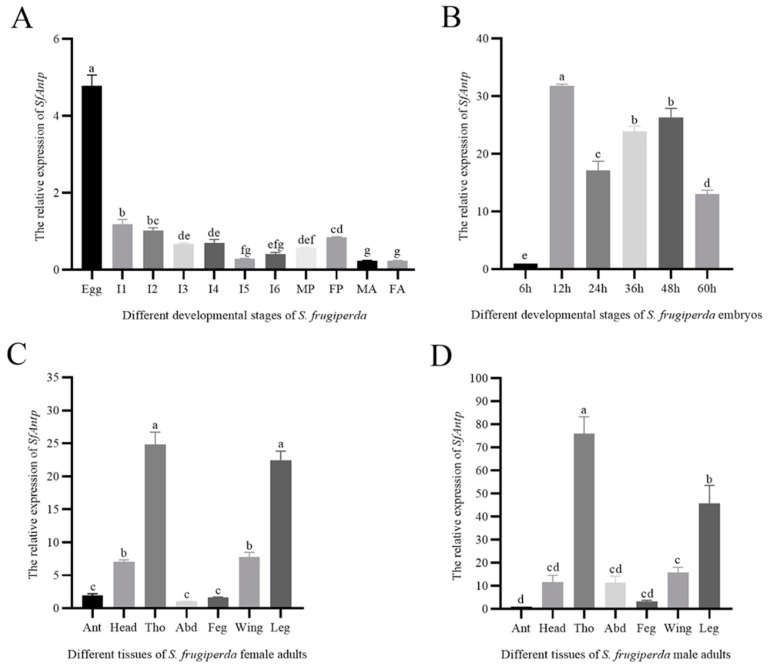
Expression analysis of *SfAntp*. (**A**) Expression profile of *SfAntp* at different developmental stages. (**B**) Expression profile of *SfAntp* in different developmental stages of embryos. (**C**) Expression profile of *SfAntp* in different tissues of female adults. (**D**) Expression profile of *SfAntp* in different tissues of male adults. I: Instar; MP: Male pupae; FP: Female pupae; MA: Male adult; FA: Female adult; Ant: Antennae; Tho: Thorax; Abd: Abdomen; Feg: Female external genitalia; Meg: Male external genitalia. The test of normality was performed using the Shapiro–Wilk method in SPSS Statistics software 26, IBM. One-way analysis of variance was used, and then Tukey’s test was used for multiple tests (*p* < 0.05). The different letters on each bar represent the significant differences between the samples. Data are expressed as the mean ± SEM.

**Figure 3 insects-15-00016-f003:**
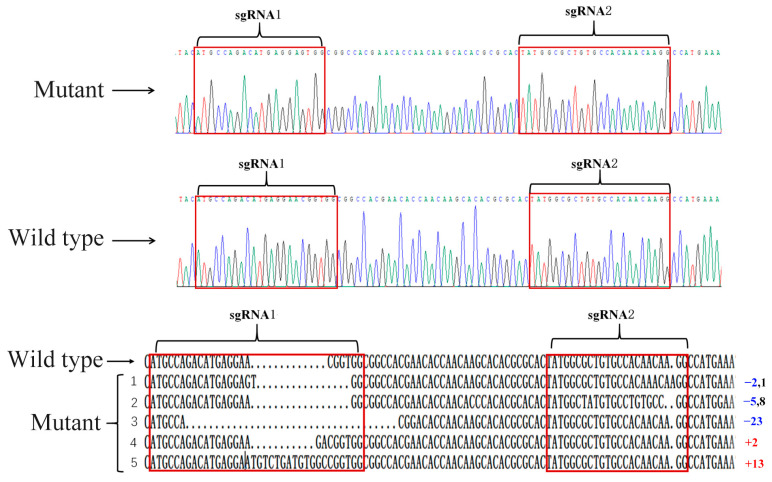
Mutant genotype detection. (**above**) Peak map of *SfAntp*. The mutant had a deletion of 3 bases at the sgRNA1 site and an insertion of 1 base at the sgRNA2 site. (**below**) Genotype of *SfAntp*. In the red box are the target sites of gene knockout. Regarding the number behind the genotype, red represents insertion, blue represents deletion, and black represents substitution.

**Figure 4 insects-15-00016-f004:**
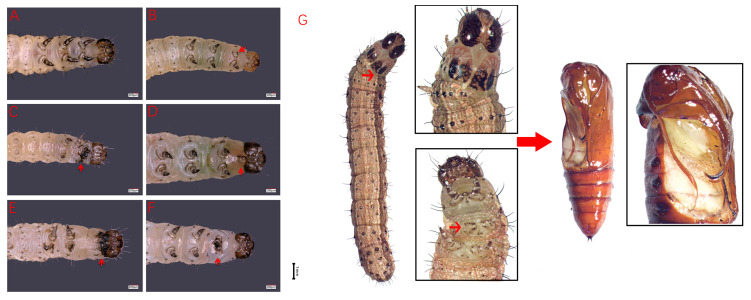
Larval phenotypes of *SfAntp* mutants induced by CRISPR/Cas9. (**A**) Wild-type larvae. (**B**) Less thoracic legs. (**C**,**F**) Thoracic leg fusion. (**D**) Leg-like projections between the head and thoracic segment. (**E**) Head and thoracic segment fusion and thoracic leg development defects. Scale bar = 100 μm. (**G**) Lack of a thoracic leg and tissue loss at the corresponding position of the thoracic leg and wing after pupation. The red arrows point to the mutation phenotype site. Scale bar = 1 mm.

**Figure 5 insects-15-00016-f005:**
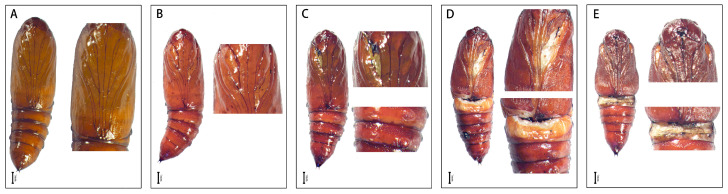
Pupal phenotypes of *SfAntp* mutants induced by CRISPR/Cas9. (**A**) Wild-type pupa. (**B**) Lack of thoracic leg corresponding position tissue. (**C**) Lack of thoracic leg corresponding position tissue, and preservation of larval leg processes in A5–A7 abdominal segment. (**D**) Lack of thoracic leg corresponding position tissue, preservation of larval leg processes in A5–A7 abdominal segment, and incision-like segments in A3–A5 abdominal segments. (**E**) The head and thoracic segments were hypoplastic, incision-like segments in A3–A5 abdominal segments. Scale bar = 1 mm.

**Figure 6 insects-15-00016-f006:**
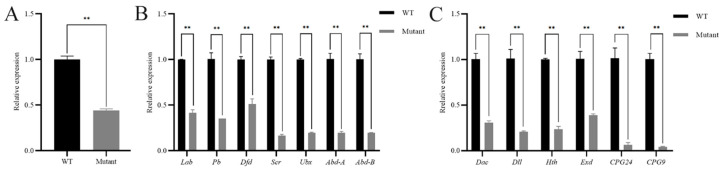
Expression changes of *SfAntp* mutant related genes compared to wild-type (WT). (**A**) Expression changes of *SfAntp* after knockout. (**B**) Expression of Hox gene family after *SfAntp* knockout. (**C**) Expression changes of four leg development related genes and two cuticular protein genes after *SfAntp* knockout. Data are expressed as the mean ± SEM of three independent technical replications. **, *p* < 0.01.

**Table 1 insects-15-00016-t001:** Primers used in this study.

Primer Name	Primer Sequence (5′-3′)	Primer Purpose
*SfAntp*-sgF-1	GAAATTAATACGACTCACTATAGATGCCAGACATGAGGAACGGGTTTTAGAGCTAGAAATAGCAAG	Preparation of sgRNA templates
*SfAntp*-sgF-2	GAAATTAATACGACTCACTATAGTATGGCGCTGTGCCACAACAGTTTTAGAGCTAGAAATAGCAAG	Preparation of sgRNA templates
Sf-sgRNA-R	AAAAGCACCGACTCGGTGCCACTTTTTCAAGTTGATAACGGACTAGCCTTATTTTAACTTGCTATTTCTAGCTCTAAAAC	Preparation of sgRNA templates
*SfAntp*-F	GATGAGCGCCAATAACTGCG	Identification of somatic mutations
*SfAntp*-R	TTTGAGATGGACTGTCGGGC	Identification of somatic mutations
*Antp*-F	ACAACCACCTCCACAACAGCC	RT-qPCR
*Antp*-R	ATCCTTCTCCTCCGCGTCA	RT-qPCR
*Lab*-F	GACTCAGGTCAAGATATGGTTCCA	RT-qPCR
*Lab*-R	CCCTTCTTTGATCCGCTTCTTT	RT-qPCR
*Pb*-F	GCGGGAGCCCCACATC	RT-qPCR
*Pb*-R	CTGTTGATAAAGCCGGTTTCG	RT-qPCR
*Dfd*-F	GACACCGCATCACCTTCACA	RT-qPCR
*Dfd*-R	AAGCAGAGCCGGGTCCAT	RT-qPCR
*Scr*	GCACAAGATGGCATCGATGA	RT-qPCR
*Scr*	GGGTGGCCGTACGGATTC	RT-qPCR
*Ubx*-F	GGGCTCAGGACTAGGTGCACTA	RT-qPCR
*Ubx*-R	TTCCTGGACTGGAGGACTCACT	RT-qPCR
*Abd-A*-F	AGTTCCACCACCAGAATTTGTTC	RT-qPCR
*Abd-A*-R	TCCCAGTCCGCCAGACA	RT-qPCR
*Abd-B*-F	TTCCTCTTTAACGCTTAC	RT-qPCR
*Abd-B*-R	AGTTCTTCTTGTTCTTCAT	RT-qPCR
*Dac*-F	GGCGGGCTGCACACA	RT-qPCR
*Dac*-R	GCACACCAGCGGCACTATG	RT-qPCR
*Dll*-F	CCAGTCTCGGCCTCACACA	RT-qPCR
*Dll*-R	ACTGCGGCGGTTTTGGA	RT-qPCR
*Hth*-F	TCCACCCCCGATGTCAGA	RT-qPCR
*Hth*-R	TCACCGCTCCACCGTATGA	RT-qPCR
*Exd*-F	CAACACGCAAGAGGAGGA	RT-qPCR
*Exd*-R	GCCAACTTCGCACGGTAG	RT-qPCR
*CPG24*	ATGAAGTTTTTGGTTGTATTGGTCG	RT-qPCR
*CPG24*	CATAAATTCCTCTCTTCTCCTGCTT	RT-qPCR
*CPG9*	TCTGGACACCATCATCAAATTG	RT-qPCR
*CPG9*	GCTGTGCTCCTTCTTCACGTAC	RT-qPCR
*GAPDH*-F	GAAGTCAAGTCCGTGGAGATG	RT-qPCR
*GAPDH*-R	GACCTGTGAAGTCG	RT-qPCR

**Table 2 insects-15-00016-t002:** CRISPR/Cas9 injection results.

Gene	Cas9 Protein/sgRNA1/sgRNA2 Concentration (ng/μL)	Number of Injections	Number of Hatching
*Antp*	600/300/300	441	67 (15.19%)
ddH_2_O	-	300	203 (67.67%)

## Data Availability

Data will be made available on request.

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
