# Peer review of "CRISPR/Cas9-Mediated Mutagenesis of Antennapedia in Spodoptera frugiperda"

_insects, 2023, doi:10.3390/insects15010016_

Round 1

Reviewer 1 Report

Comments and Suggestions for Authors

In this work, the authors investigated the expression and function of one of the developmentally important Hox gene, Antennapedia (Antp), in a well-known agricultural pest, the fall armyworm Spodoptera frugiperda. Using RT-qPCR they characterized the stage and tissue-specific expression of the SfAntp gene, showing the highest expression during embryogenesis and, in agreement with literature data, in thoracic segments and legs. Then they knocked out the SfAntp gene using CRISPR/Cas9 genome editing method and a few mutant larvae and pupae, which survived the treatment, demonstrated the function of this gene by developmental abnormalities. The results are novel and compelling and support the conclusions drawn, making the study worth publishing.

However, the manuscript is very poorly prepared. The first problem is the English. I have made a large number of minor suggestions to correct some errors (but certainly not all). The manuscript needs a thorough language revision, either by a native English speaker or by an English editing service. The second problem is a scientific style (see my specific comments). The manuscript contains a number of shortcomings and some inaccurate claims, the methods used are superficially described (some even missing) and some illustrations have incomplete legends or even errors in the legend (see my comment to Figure 1C). These shortcomings significantly diminish the scientific value of this work. In other words, this manuscript requires thorough revision to be acceptable for publication.

Specific comments

(1) Abstract, L23-24: As it is written now, the meaning of the sentence beginning with “Two sgRNA sites” is wrong (nothing can be knocked out by microinjection only). Perhaps, the sentence could be reworded as follows: “Two sgRNA sites were designed at the first exon of SfAntp and the gene was knocked out by CRISPR/Cas9 via microinjection.”

(2) Introduction, L64-66: Authors refer to Meng et al. (2015) when describing the effect of the Antp absence in Bombyx mori. However, no such information is given in the publication of Meng et al. (2015). This publication is about the regulation of transcription of the Phantom gene encoding a steroidogenic enzyme required for the synthesis of ecdysone. This reference should be replaced with a proper reference.

(3) Introduction, L66: Explain Wes mutation, for example, “The Wedge eye-spot (Wes) mutation”. In addition “Wes” should be written in italics (also L71).

(4) Materials and Methods: I completely lack information about the origin of the laboratory population of Spodoptera frugiperda that was used for these experiments, as well as information about diet and rearing conditions. This information must be provided in this section.

(5) Materials and Methods, 2.1. Analysis of SfAntp expression profile, L99-101: when determining time after oviposition, it is better to use terminology usually used such as “hours post oviposition (hpo)” instead of the incorrect term “after delivery”. Then the text in brackets would read “[40-50 eggs were collected at 6 hours post oviposition (hpo), 12 hpo, 24 hpo, 36 hpo, 48 hpo and 60 hpo, respectively]”.

(6) Materials and Methods, 2.2. Phenotypic changes caused by SfAntp mutation, L120-130: There is no information about the age of eggs used for microinjections. I am also missing details about the microinjection procedure, such as the type of microinjector and the type of injection needles (also thickness, with or without filament). These are important factors because the hatchability of injected eggs was relatively high in the control group (as described in Results, 3.3.). I also miss information about the method used for DNA extraction from mutant larvae and pupae (see L128-129.

(7) Materials and Methods: The subsection on the phylogenetic analysis of SfAntp is missing.

(8) Results, 3.1. 3.1. Phylogenetic analysis of SfAntp, L141: Pieris chrysantha? I do not know such species of Pieris butterflies! Should not be here Pieris rapae, which is given in Figure 1?

(9) Figure 1C legend: Several GenBank accession numbers of the amino acid sequence of Antp given in this legend are wrong. Namely, the acc. no. of S. frugiperda (XP_035446614.1) and S. litura (XP_022820536.1) are numbers of Wnt-1 proteins in these species and the acc. no. of B. mori (NP_001166802.1) is the number of the abdominal B isoform RBS. Please provide the correct accession numbers! Also, provide information about what the numbers in the phylogenetic branches mean and what the scale bar shows.

(10) Results, 3.3. Phenotypic changes caused by SfAntp mutation, L195: What is monoclonal sequencing? That should be explained.

(11) Figure 4A-F: Red letters in the left low corner, indicating individual pictures, are two small. Make them bigger.    

(12) Results, 3.4. Expression changes of SfAntp mutant related genes, L234-L243: I miss a reference to Figure 6A in this subsection.

(13) Discussion: The phylogenetic analysis of Antp done by the authors in S. frugiperda and several other representatives of insects is not discussed in the Discussion. In the constructed tree, five Lepidoptera species are clustered together, as well as three Diptera species. However, two beetles (Coleoptera) show a closer relationship with the brown planthopper Nilaparvata lugens, representing a very distant insect order Hemiptera, than with two Endopterygota orders, Lepidoptera and Diptera. 

(14) Discussion, L287-288: The authors refer to Xin (2015) for the effect of Antp mutation on the development of wings in the silkworm, Bombyx mori. However, I was unable to find any reference to Xin's (2015) publication by searching the internet. In addition, the citation of this publication in References is incomplete (see reference no. 32, L382).

(15) Conclusions, L308-309: The two following sentences are redundant and can be omitted (this information is already provided in L302-305): “In this study, it was confirmed that Antp also plays an important role in S. frugiperda. At the same time, it also affects the expression of other Hox genes.”

Minor suggestions

L2: omit “the” before Spodoptera frugiperda in the title

L16: ... the Antp of S. frugiperda (SfAntp) ...

L18: the phrase “and the results that it was expressed throughout the life cycle of fall armyworm” is not understandable due to incorrect English, and it should be reworded

L19 and L20: make space between the number and the unit (h), i.e. “6 h”, “12 h”, “48 h”

L26/27: In addition, deletion of SfAntp resulted in high embryo mortality.

L30: Spodoptera frugiperda;

L34: Spodoptera frugiperda (J. E. Smith)

L37: Alton and Sparks, 1979

L41: (Stokstad, 2017; Sharanabasappa et al., 2018; Jiang et al., 2019)

L77: … in the spider Achaearanea tepidariorum, …

L91-92: (Ronshaugen et al., 2002; Di-Poi et al., 2010; Mansfield and Abzhanov, 2010)

L93: ... CRISPR/Cas9 mutagenesis of the S. frugiperda Antp (SfAntp) gene.

L94-95: ... of this Hox gene in S. frugiperda,

L104-106: ... an appropriate number of heads, thoraxes, abdomens, external genitalia, legs and antennae of adult females and males.

L109: ... designed by Primer software.

L111-116: The content of each reaction was: 10 μL of Power UpTM SYBRTM Green Master Mix, 1 µg of cDNA, 1 μL of 10 μM forward and reverse primers each, plus RNase Free ddH2O to 20 μL. The reaction conditions were: 50 °C for 2 min, 95 °C for 2 min; 40 × ( 95 °C for 15 s, 60 °C for 1 min); 95 °C for 15 s, 60 °C for 1 min to 95 °C in 0.15 °C steps of 1 s for the dissolution curve.

L115: ... a reference gene, ...

L117: Student's t-test [“t” should be written in italics]

L118: P < 0.05 [“P” should be written in italics]

L122: (300 ng/μL)

L123: (300 ng/μL) and Cas9 (600 ng/μL)

L124: ddH2O [“2” should be written as subscript]

L124: as a control group.

L125: a Petri dish

L126: the Petri dish [2x]

L133-134: We searched NCBI for sequences of other Hox genes (...) and used Primer ...

L152: (XP_038217532.1);

L157: SfAntp” should be written in italics

L167: at 6 hpo

L168: at 12-48 hpo

L170: Figure 2. Expression analysis of SfAntp.

L175: (P<0.05) [“P” should be written in italics]

L177: SfAntp” should be written in italics

L204: ... 3 pairs of thoracic legs on T1-T3 (Figure 4A), ...

L205: After SfAntp was ...

L207: ... less thoracic legs, ...

L222: Figure 4. Larval phenotypes of SfAntp mutants ...

L222: (A) Wild-type

L223: (B) Less thoracic legs.

L224: and thoracic segment. (E) ...

L228: Figure 5. Pupal phenotypes of SfAntp mutants ...

L228-229: (A) Wild-type pupa.

L237: After SfAntp mutation, Hox genes were down-regulated ...

L239: ... the effect of SfAntp on ...

L245: Figure 6. Expression changes of SfAntp mutant related genes compared to wild type (WT).

L248: **, P < 0.01.

L259: 12 h and 48 h

L260: SfAntp

L262: thoracic segments and legs

L277: … is relatively conserved.

L285: … to be conserved.

L303 and L306: SfAntp

L305: … of other Hox genes in S. frugiperda.

L343-344: Sharanabasappa, Kalleshwaraswamy, C.M, Asokan, R., Mahadeva Swamy, H.M., Maruthi, M.S., Pavithra, H.B, Hegbe, K., Navi, S., Prabhu, S.T., Goergen, G., 2018.

L364: Hox proteins in the regulation of muscle development.

L375-376: Proceedings of the National Academy of Sciences of the United States of America

L377-378: A Hox transcription factor collective binds a highly conserved Distal-less cis-regulatory module to generate robust transcriptional outcomes.

Comments on the Quality of English Language

The manuscript needs a thorough language revision, either by a native English speaker or by an English editing service.

Reviewer 2 Report

Comments and Suggestions for Authors

CRISPR/Cas9-Mediated Mutagenesis of Antennapedia in the Spodoptera frugiperda

insects-2728813

In this study, Wang et al. cloned the Antennapedia gene, studied the sequence by blasting and phylogenetic tree construction, and then measured its spatio-temporal expression by using qPCR in S. frugiperda. They then knocked out this gene by using the CRISPR/Cas9 system and studied the morphological changes in mutants, and further measured the expression of Hox gene family in mutants. In its present form the MS is highly unacceptable.

Major comments:

A ‘Simple Summary’ is required for this journal.

The abstract needs to be rewritten to cover the main findings.

Introduction:

A deeper and extensive review is required, such as the expression of the gene, its functional mechanisms, not only in Drosophila and moths, and may be the CRISPR/Cas9 system used in moths or Spodoptera.

Materials and Methods:

This part need to be substantially improved, particularly the methods information. Detailed research and analysis methods, including insets rearing, life cycle, statistics, injection time and amount, phylogenetic tree construction, etc., need to be addressed very carefully.

For the qPCR, you used the 2-△△Ct method outlined by Livak-Schmittgen for calculating the relative expression levels. This method based on the assumed efficiency of 100% for all target and reference genes. Please provide this information in the M&M if you had performed such analysis and also the data for review.

Statistics methods and reporting:

Please see my comments in the PDF file.

Results:

Some contents should be moved to M&M.

Discussion:

The results are not discussed sufficiently up against what other people have found.

Please see other detailed or minor comments in the PDF file.

Comments on the Quality of English Language

Throughout the MS the English language should be improved. Many companies are available for that and a native English speaking scientist close to the research field should also proofread the MS. 

Round 2

Reviewer 1 Report

Comments and Suggestions for Authors

The revision has been very well done. All my comments and suggestions have been satisfactorily addressed and explained in the response letter. The manuscript is now acceptable for publication after a few minor text edits as outlined below.

Minor suggestions

L47: delete “are distributed”

L49: replace “Kalleshwaraswamy et al., 2018” with “Sharanabasappa et al., 2018”

L139: using the

L147: required

L150: it was necessary

L151: at 6 h, 12 h, 24 h, 36 h, 48 h and 60 h after oviposition

L152: required

L162: 1 µg of cDNA

L179: Petri dish

L208: the authors use the abbreviation as suggested, but forgot to define it – please do it as follows: “at 6 hours post oviposition (hpo)”

L242: wild-type

L256: leg-like

L267: pupa

L325: lepidopteran

L327: lepidopteran

L423: International Journal of Molecular Sciences

L444: Science 356, 473-474.

L448: PLoS Genetics

Comments on the Quality of English Language

English is now fine, only minor edits required, as specified in my review.

Reviewer 2 Report

Comments and Suggestions for Authors

1.

*Figure 2: Detailed statistical method is required. suggest using ANOVA followed by Tukey's studentized range test for multiple comparisons for normal data. while for not normal data, using the nonparametric Kruskal-Wallis test followed by Dunn’s procedure with Bonferroni correction for multiple comparisons. also the statistical value, df and p value for the whole model should be provided.

Response:

Thank you for your scientific attitude and helpful comments.

The data were analyzed by single factor analysis of DPS software, then Dunn-Sidak method was used for multiple comparisons.

**The statistical methods still not clear, such as what single factor analysis, the software, normality test. The authors may need to consult a statistician for this issue.

2.

*For the qPCR, you used the 2-△△Ct method outlined by Livak-Schmittgen for calculating the relative expression levels. This method based on the assumed efficiency of 100% for all target and reference genes. Please provide this information in the M&M if you had performed such analysis and also the data for review.

Response:

Thank you for your scientific attitude and helpful comments.

Because we used a single internal reference gene analysis, the formula itself default primer amplification efficiency is 100%, so we did not calculate the primer amplification efficiency one by one.

**The efficiency generally needs to be tested to fit the requirement of the Livak-Schmittgen’s method.

3.

**L125-138. Move the species’ name and ID to the tree.

4.

**The conclusion is still too long.

5. 

L126. "litura(GenBank)"  to "litura (GenBank)"

6. 

L139-145. Not clear and showed some repetitions.

Comments on the Quality of English Language

Moderate editing of English language required
